**Data Availability Statement:** All relevant data are within the paper and its Supporting Information files. Fully de-identified interview transcriptions are also available should they be needed.

# A qualitative study to explore the experience of parents of newborns admitted to neonatal care unit in rural Rwanda

Samuel Byiringiro[1,2,¤a]*, Rex Wong[2,3], Jenae Logan[2,¤b], Deogratias Kaneza[4], Joseph Gitera[5], Sharon Umutesi[6], Catherine M. Kirk[1,2]

1 Maternal and Child Health Department, Partners In Health/Inshuti Mu Buzima, Rwinkwavu, Rwanda, 2 Bill and Joyce Cummings Institute of Global Health, University of Global Health Equity, Butaro, Rwanda, 3 School of Public Health, Yale University, New Haven, Connecticut, United States of America, 4 Ruli District Hospital, Ministry of Health, Gakenke, Rwanda, 5 Clinical and Public Health Services Division, Ministry of Health, Kigali, Rwanda, 6 Maternal, Child and Community Health Division, Rwanda Biomedical Center, Kigali, Rwanda

¤a Current address: Johns Hopkins University School of Nursing, Baltimore, Maryland, United States of America
¤b Current address: Partners In Health, Boston, Massachusetts, United States of America
* sam6as@yahoo.com

## Abstract

### Background

Neonatal Care Units (NCUs) provide special care to sick and small newborns and help reduce neonatal mortality. For parents, having a hospitalized newborn can be a traumatic experience. In sub-Saharan Africa, there is limited literature about the parents' experience in NCUs.

### Objective

Our study aimed to explore the experience of parents in the NCU of a rural district hospital in Rwanda.

### Methods

A qualitative study was conducted with parents whose newborns were hospitalized in the Ruli District Hospital NCU from September 2018 to January 2019. Interviews were conducted using a semi-structured guide in the participants' homes by trained data collectors. Data were transcribed, translated, and then coded using a structured code book. All data were organized using Dedoose software for analysis.

### Results

Twenty-one interviews were conducted primarily with mothers (90.5%, n = 19) among newborns who were most often discharged home alive (90.5%, n = 19). Four themes emerged from the interviews. These were the parental adaptation to having a sick neonate in NCU,

**Funding:** CMK: Received funding from Grand Challenges Canada Saving Lives at Birth as part of the evaluation of the All Babies Count initiative. SB: University of Global Health Equity covered the cost of Dedoose software utilized in the analysis. The funders had no role in study design, data collection and analysis, decision to publish, or preparation of the manuscript.

**Competing interests:** The authors have declared that no competing interests exist.

**Abbreviations:** ABC, All Babies Count; LMIC, Low- and Middle-Income Countries; NCU, Neonatal Care Unit; RNEC, Rwanda National Ethics Committee; PTSD, Post-Traumatic Stress Disorder.

adaptation to the NCU environment, interaction with people (healthcare providers and fellow parents) in the NCU, and financial stressors.

## Conclusion

The admission of a newborn to the NCU is a source of stress for parents and caregivers in rural Rwanda, however, there were several positive aspects which helped mothers adapt to the NCU. The experience in the NCU can be improved when healthcare providers communicate and explain the newborn's status to the parents and actively involve them in the care of their newborn. Expanding the NCU access for families, encouraging peer support, and ensuring financial accessibility for neonatal care services could contribute to improved experiences for parents and families in general.

## Background

Neonatal mortality makes up nearly half of all deaths among children under five years of age [1]. It is estimated that nearly 1.9 million newborn deaths can be averted by 2025 if quality maternal and newborn care interventions were implemented [2]. Expansion of access to specialized neonatal care units (NCUs) for sick and small newborns is one of the global priorities to reduce neonatal mortality [1]. Despite the tremendous benefits of NCUs in achieving better neonatal health outcomes, studies exploring the experience of parents in NCUs have shown it to be a stressful experience for families [3–8].

While there is a large body of literature regarding parental experience in NCUs in western countries, very little is known in Sub-Saharan Africa [9, 10]. A small number of studies in South Africa showed that parents often experience negative emotional feelings and various challenges in NCUs [8, 11]. Many parents expressed depressive symptoms and inability to bond with their newborns [12–16]. Some newborns were abandoned in the NCU, and this disruption of parental care can contribute to inferior early child development outcomes, lower educational attainment, and poorer lifetime economic earnings [17]. However, the findings and recommendations from South Africa—a high middle-income country, are difficult to transfer to other Sub-Saharan African nations—predominantly low-income countries [18].

Between 2005 and 2015, Rwanda has documented a rapid decline in under-five child mortality from 152 to 50 deaths per 1,000 live births and neonatal mortality from 37 to 20 deaths per 1,000 live births [19]. Despite such tremendous progress, neonatal deaths still contribute to 40% of all under-five deaths in Rwanda [19]. In order to accelerate the reduction in neonatal mortality, the Rwandan Ministry of Health launched the National Neonatal Care Protocol in 2012 addressing the gaps directly related to clinical neonatal care [20] and scaled-up hospital level neonatal care countrywide. However, other than for skin to skin care, the protocol does not integrate parents into the NCU environment [21, 22]. One quantitative study found high stress score and the predominant stressors were the babies' appearance and behavior among parents of newborns admitted to NCU in an urban tertiary hospital [23].

Quality healthcare requires both the provision of evidence-based clinical care to newborns and also ensuring a positive experience of the newborn and their families [24]. High quality care cannot be fully achieved unless the experience of care meets the needs of families, which requires knowing their experiences, and designing appropriate interventions [24–26]. Accordingly, this study explored the experience of parents whose neonates were hospitalized and

discharged from the NCU in a district hospital in rural Rwanda. The results of this study can add to the existing literature on the experience of parents with newborn children in the NCU in low-and middle-income countries in the sub-Saharan Africa region.

## Methods

### Setting

This study was conducted in the catchment area of Ruli District Hospital–a rural public hospital located in Rwanda's Northern Province. With 10 beds and 5 incubators, Ruli District Hospital's NCU also serves as a referral site for hydrocephalus surgical treatment. In 2017, there were 459 NCU admissions [27]. From 2017–2019, Ruli District Hospital received support to improve the quality of care for newborns through the All-Babies Count (ABC) program implemented by Partners In Health/Inshuti Mu Buzima and the Ministry of Health.

**NCUs at District hospital levels in Rwanda.** NCUs at the district level in Rwanda provide preventive, curative, and rehabilitation care to the small and sick newborns, and those at risk for complications [28, 29]. The majority of babies who are referred to the district level NCUs are usually those born too early and/or too small (with weight below 2.5 kg), but occasionally too big (above 4 kg); babies with birth asphyxia; and babies with risk for infection as stated in the national clinical guideline (including but not limited to babies born at home, from mothers who presented with a fever during labor or 24 hours after delivery, or rupture of membranes 18 hours before delivery) [28, 30].

District level NCUs exist as a separate unit from maternity even though they work closely together. They treat newborns from both the health centers of the hospitals' catchment area and the hospital maternity. Babies born at home are automatically referred to the NCU for risk of infection, and while those born at the health centers and the hospital maternities are evaluated for any necessity to refer to them to the NCU. If transferred to the NCU, hospitals' procedures vary but at Ruli DH, the baby's mother or closest relative accompany the baby inside the NCU [28].

NCUs in Rwanda provide care at varying capacities yet the health service package for public health facilities determines that they should be able to offer supportive oxygen to babies with birth asphyxia, continuous positive airway pressure to premature and term babies with respiratory distress, antibiotic therapy, thermal regulation support, fluid and electrolytes when necessary to name a few [29]. Specialized neonatal care is usually absent at district levels and newborns who require it are sent to referral hospitals' NCUs.

### Design

We used qualitative descriptive and contextual approach. Semi- structured interviews were conducted to understand the experiences of parents having a newborn hospitalized and discharged from the NCU.

### Study participants

Parents or primary caregivers of neonates discharged from the NCU between September 2018 and January 2019 (three months prior to data collection) were included in the study. To ensure that parents and caregivers had been amply exposed to the NCU experience, we only included those whose newborns were admitted to the NCU for at least three days. While some mothers and caregivers were aged below 18 years, we excluded them since their experience tends to be different because of stigma associated with teenage pregnancy in Rwanda [31]. The acquisition of consent for the participants aged below 18 is additionally different. To ensure diversity and

a mix of experiences, we purposively stratified the samples by distance (living within and more than 1-hour walking distance from the facility), by admitting diagnosis (prematurity and other reasons), and outcome (died or alive at discharge).

## Recruitment procedure

The recruitment took place between December 2018 and January 2019. We made the list of eligible potential participants and their geographical locations from the Ruli District Hospital NCU registry. We then contacted the nearby Community Health Worker (CHW) to pass on the recruitment message on our behalf and book us an appointment to the potential participants. On the appointment date, the interviewer was accompanied by the CHW to the potential participant's home which is where the data collection took place.

## Data collection tool and procedures

Interviews were conducted using a semi-structured interview guide that included questions related to the parents' and caregivers' feelings on neonatal admission and their experience in the NCU, the quality of communication with healthcare providers, their involvement in the newborn's care, and the support received from the hospital staff, relatives, or others during the stay. We developed the interview guide in English and translated it to Kinyarwanda—the local language for conducting the interviews.

The interview guide was pre-tested and piloted on two parents before the actual data collection. The pilot did not lead to changes in the protocol hence its participants were included in the analysis. Two female data collectors who were not healthcare providers nor worked in the study district were trained to conduct the interviews. Husbands and wives were interviewed separately. The interviews were audio recorded and took place in a quiet place at the participants' homes. The study participants received a gift consisting of a baby hat and a bar of soap. The Principal Investigator listened to the interviews and reviewed notes at the end of each data collection day to determine if the saturation was reached, adjust the probes as needed, and/or identify any areas of improvement in the way the interviews were being moderated. To further ensure the trustworthiness of the study and its findings, and better conduct of the research, the team met every end of the day of data collection to debrief on the data collection experience.

## Data analysis

The interviews were transcribed and translated, then coded by two independent investigators. We explored the literature on the experience of parents in typical low-income settings' NCUs and used it to create the preliminary codebook. The codebook was amended using pilot interviews. We further ensured rigor and trustworthiness in the analysis. The data collectors were trained on data coding since they were very familiar with the data. The coders worked independently and met with the Principal Investigator to resolve any discrepancies by discussion and revision of the definition of the codes. We used Dedoose software to organize data for analysis. The coded transcripts were grouped into themes and we chose the representative quotes to include in the findings.

## Ethics

The study was approved by the Rwanda National Ethics Committee (No. 107/RNEC/2019) and the Institutional Review Board for the University of Global Health Equity. Ruli District Hospital provided approval for NCU data access. The potential participants were provided information about the study and explained that participation was voluntary. Those who accepted to take

part in the study provided written informed consent before the interview begins. To protect the participants' confidentiality, we utilized participant numbers in the transcripts instead of their names, and we de-identified all quotes included in the results by removing details that could lead to the recognition of the individual participant or their healthcare provider.

## Results

A total of 21 interviews of 45 minutes in average each were conducted (Table 1). Participants were aged between 20 and 56. Nineteen (90.5%) were female participants. Seventeen (80.9%) of the participants were married and 4 (19.0%) were single. Nineteen (90.5%) had children who were alive, while 2 (9.5%) died prior to discharge from NCU. Eight (31.0%) lived within an hour walking distance to the hospital while 13 (61.9%) had to either take the bus or a motorcycle or walk for more than an hour to the hospital from their homes. Prematurity (42.8%) and infection risk or fever (14.2%) were the most common reasons for admission. All participants had health insurance.

Four overarching themes were generated from the interviews with parents regarding their experience of having their newborn hospitalized in NCU at Ruli DH. Theme one consists of parental emotional adaptation to the newborn care in the NCU. The second theme describes the parental adaptation to the NCU environment which includes the physical environment and rules and regulations of the NCU. The third theme covers the experience of parents with other people in the NCU and the last theme discusses the Financial aspects of the care and support received at Ruli DH.

### Theme 1. Parents' emotional adaptation to the newborn care in the NCU

Parents' emotional adaptation to receiving newborn care in the NCU was first characterized by a mix of negative feelings and emotions resulting from the quick roll out of events in the

**Table 1. Demographic characteristics of participants.**

| Characteristics | | N (%) |
|---|---|---|
| Total Participants | | 21 |
| Age (n = 19) | 20–29 | 8 (38%) |
| | 30–39 | 8 (38%) |
| | > = 40 | 3 (15%) |
| | Undeclared | 2 (9%) |
| Gender | Male | 2 (9.5%) |
| | Female | 19 (90.5%) |
| Newborn's outcome at discharge | Alive | 19 (90.5%) |
| | Dead | 2 (9.5%) |
| Marital status | Married | 17 (80.9%) |
| | Single | 4 (19.0%) |
| Distance to hospital | <1 hour | 8 (38.0%) |
| | >1 hour | 13 (61.9%) |
| Reason for admission | Prematurity | 9 (42.8%) |
| | Low Birth Weight | 2 (9.5%) |
| | Infection risk/Fever | 3 (14.2%) |
| | Other* | 7 (33.3%) |
| Health Insurance | Yes | 21 (100%) |

[a]Other: asphyxia, congenital malformation, jaundice, not sucking on breast, respiratory distress, cord bleeding.

short time from stress of delivery to the admission in the NCU. The parents found the NCU environment to look strangely different because of the unusual equipment and the type of care newborns receive. They feared that the baby was being hurt by the different nursing and medical care and procedures.

**1.1. Stress due to baby's condition.** Many parents reported feeling fearful when they were informed that their newborn required admission to the NCU. They feared that the child could die and that caused them sadness, uneasiness, hopelessness, disturbance, and anxiety.

*"There was no other way I could understand it. . . I didn't even feel hopeful because the baby was too premature. I was just waiting for what God was to give me. My baby only weighed 1.2 kg. I felt hopeless and felt the healthcare providers were struggling for nothing."* (Mother, age 36).

Several mothers reported a very emotional response to their newborn's admission:

*"I felt very sad and I am telling you that I would suddenly burst into tears. I was alone with the baby always expecting to be discharged the following day. I felt disturbed, wept repeatedly, then breastfeed the baby but suddenly found myself crying."* (Mother, age 28).

*"I cried. Other mothers used to say that no other person enters there [NCU]. I felt like it was over for me when I saw myself referred to there. I thought that I would not leave soon and I imagined that my baby would eventually die. I felt very sad, and it was hard for me to accept the situation"* (Mother, age 25).

**1.2. Stress due to the visual sight of the baby and the NCU.** Many parents reported negative feelings caused by the look at the different tubes and wires attached to the newborns inside the incubators, the sound of the equipment and sometimes the care provided to the newborns. While the fathers were not allowed into the incubator room part of the NCU, one father reported managing to trespass into that room and described how he felt when he saw his fifth newborn in the incubator for prematurity:

*"I felt scared to see the machines. You say, look at these machines, and the baby is lying in them naked. I was not sure that my baby would survive."* (Father, age unknown).

Another mother shared:

*"I was stressed because I was thinking that she (my baby) would not survive. I saw that my baby was on oxygen, and another noisy machine, I lost hope because the baby spent the whole week in the incubator. When I arrived at the hospital, I was so worried that my baby would not make it."* (Mother, age 35).

**1.3. Stress due to that fear the baby is being hurt during care.** Additionally, parents feared that their newborn may be hurt by the care provided in the NCU. The fear was particularly higher when the parents thought the provider was not experienced enough (student for instance) to care for the sick baby.

*"He [a nursing student] held it [a needle] and pierced without first looking for where the vein was, and he attempted three times. The third time, I told him to stop paining my baby."* (Mother, 32).

## Theme 2. Parental adaptation to the NCU environment

Parents reported some regulations that were positive–such as uninterrupted access to their newborn but other regulations of no visitors, no caregivers, and no eating inside the ward stressed mothers. The procedures of always cleaning hands, changing shoes, and putting on different clothing (aprons) are a source of stress to some mothers even though some mothers appreciated them due to the resulting calmness in NCU. Caregiver's stress was worsened by insufficient induction to the NCU contributing to a lack of understanding of NCU rules and procedures.

**2.1. Being part of the baby's care.** Mothers reported that having full access to the newborn any time throughout hospitalization, even when the newborns were inside incubators or on phototherapy, brought them comfort. Having some responsibility for the care of their newborns, ranging from paying the bills, to bringing required materials, to feeding or bathing their newborn, improved their feelings about the situation even though the part of responsibility was occasionally a source of stress to parents who were unable to bring fulfill all their responsibilities (Paying bills, providing newborn materials such as clothes, bringing hygiene materials and so on).

> *"The healthcare providers told us, we could go at any time to have a look at our babies. They told us to be close to our babies. We hence used to come look at our babies and [go] back whenever."* (mother, age 35).

> *"They [nurses] told me to go and breastfeed my baby anytime I wanted. I felt no problem because I was free to go and see the baby anytime."* (mother, age 22).

The mothers rejoiced in being able to access their newborns at any time, but the inability to care for their newborns such as lacking breast milk was the source of stress. The stay was additionally unpleasant for some mothers who lacked means to follow up their own health because they were still sick when the newborns were transferred to the NCUs.

> *"What stressed me is when they told me to express breastmilk for my baby. The baby stayed in the machines on oxygen, and when it was time to start pumping breast milk for the baby, I could not get it."* (mother, age 36).

> *"Because of taking care of the baby while I was still sick, I became sicker and felt as if I was dead. I did not know what I was sick of but when I went to seek care I had high fever and was shivering."* (mother, age 33).

**2.2. Access to their babies.** The fathers were the only caregivers other than the mothers reported to have some access into the general room (Kangaroo Mother Care Unit), though they had limited visiting hours and were not allowed to touch or hold the newborns, or see them if they were in the incubators' room. Other visitors were not allowed into the NCU. The mothers felt very sad when the newborn passed away without having the father see them.

A mother reported her feeling that the father did not see their newborn before she died:

> *"I told one of the other mothers in the NCU that I was very sad because my baby was going to die before her father could at least see her. The mother told me that I should have requested it before, and I told her that nobody advised me to ask a healthcare provider for permission so that my husband could visit my baby."* (mother, age 35)

**2.3. Orientation to the NCU.** Parents did not appreciate the fact that they were not provided a rigorous initial orientation so that they consciously abide by the NCU regulations. They would find themselves frustrated by the things about which they should have been informed at admission. Most of quarrels between parents and healthcare providers resulted from the tendency of mothers to bring visitors to the NCU.

*"She [healthcare provider] asked me why I had violated the rules [allowing the father of the baby to visit]. She replied that I should not allow him in without telling her first. I told her that I had read the regulation but did not know that I should get approval from her before bringing him inside."* (Mother, age unknown).

*"They [healthcare providers] should explain to the mother where they were taking the baby and explain the life there. For instance, after I arrived, nobody explained the rules and regulations of the place except reading it for myself. None told me the procedure of things in the unit. They should improve that."* (Mother, age 25).

**2.4. Physical environment of NCU.** In addition to the stressors from the NCU regulations, mothers reported the NCU interior to have low temperature which make it chilly. They even feared that the coldness they felt affected the babies as well. In the NCU, the mothers had a responsibility to look after their babies, day and night ensuring their cleanliness, nutrition, changing diapers and clothes. . . These responsibilities made the mothers unable to rest adequately and it was an additional challenging experience for them.

*"Cold. . . ohhhh, it was too cold. Our babies were freezing and were losing weight. The room was freezing."* (mother, age 44).

Despite the many stressful aspects of the NCU, mothers reported the environment to be safe from disorder and robbery, clean and quiet which provides a sense of comfort and positive feelings. Some mothers reported appreciating the safeguard put in place to minimize visitations explaining that such measures allow the NCU environment to remain clean, and silent which they thought is good for the sick newborns.

*"I thought the place would be smelling bad. I, however, found it different. They [healthcare providers] teach them [mothers] morning and evening to have hygiene so that even newcomers can be informed about hygiene."* (mother, age 43).

*"I was very happy with the hygiene of the NCU. I was happy with the way they restrict the access into the Unit where there are babies born with problems. You see, allowing many people inside would cause the room of sick babies to be stuffy. They avoided that. The way they prohibit the entry of shoes from outside is good."* (mother, age 27)

*"You enter and you can even make a prayer silently in your mind to avoid making noise. You make sure there is no noise in the NCU, no one is allowed to scream, the people inside are so quiet. It is not authorized to make phone calls or other devices that may produce noise."* (mother, 35).

## Theme 3. Experiences with other people in NCU

Parents reported having both positive and negative interactions with healthcare providers, and an overall strong sense of trust for healthcare providers. Other mothers in the NCU were reported to be a consistent source of support (emotional and material) for each other.

**3.1. Interactions with healthcare providers.**   A mix of healthcare provider attitudes, some positive and others negative, were reported by parents. They were appreciative of the healthcare providers who explained the care their newborns received, provided guidance and updates on the newborn's condition, and encouraged them to ask questions.

*"There was one woman [healthcare provider] who used to talk to us. She would say, 'Do you have any questions?', and we often responded, 'no'. She would then ask us what she would have taught us. We were more comfortable with that healthcare provider."* (Mother, age 28).

*"We would joke with healthcare providers and occasionally forget that our babies are hospitalized. They would come and comfort us telling us that our babies would get cured. You could notice that they were doing whatever possible to converse with us."* (Mother, age 33)

The provision of information about the findings of assessments contributed to the parent's understanding of the newborn's condition and was one key factor to the positive feelings and comfort to the mothers in the otherwise hopeless conditions.

*"It was my first time to see such a disease [jaundice], my first time to hear of it. I had hoped that he would get well soon. After they [healthcare providers] explained the condition to me, I felt ensured that the child shall get cured and I had no problem."* (Mother, age 22).

*"When I arrived, they [healthcare providers] examined the baby, looked at her vomiting and the problem in her eyes and they comforted me saying it would go away little by little. They advised me how to position my baby while breastfeeding and it decreased gradually. Gradually as they talked to me and comforted me, I regained the mood."* (Mother, age 28).

However, some healthcare providers were reported to have a bad attitude which made the NCU stay stressful and challenging for some mothers. Parents reported a negative experience of interacting with some healthcare providers who spoke in a rude tone, provided no guidance or information, and showed no empathy while requesting newborn care materials from parents.

*"Whenever healthcare providers entered, I trembled because they entered speaking roughly, except that you ought to get used to it. Once, I went outside immediately after breastfeeding. A few minutes after, when I came back the baby had passed stools. The healthcare provider immediately talked to me in a rude voice. I tried to explain that I was there with my baby a few seconds before. She refused to understand and roughly said I should look after my baby at every moment."* (Mother, age 22).

*"They yelled at me asking me why I had not yet bought clothes for my baby. I begged and told them that I could not afford to buy the clothes because I am poor. She rudely replied that poor people do not go to seek care at Ruli hospital."* (Mother, age 36).

Blaming parents for the newborn's condition was another attitude repeatedly reported by parents. In most cases, the parents reported that healthcare providers blamed them for not taking good care of their babies or not fulfilling their responsibilities (such as providing needed materials for newborn care) hence leading to worsening of the condition.

*"My baby had a fever and was vomiting. I called the healthcare provider who was on day duty and she said that it was because of me that I was not holding my baby well."* (Mother, age 22).

Most of the parents, including those whose newborns died prior to discharge, trusted that providers were doing their best to save the newborns.

*"I did not decline any treatment because I could see them coming to treat my baby and I felt that because they are healthcare providers my baby would be cured."* (Mother, age 33).

*"I felt that because my baby is in the hands of healthcare providers, she would be fine. They would comfort us saying that no baby would die anymore, and we felt hopeful. After that, they took an exam and the result was good and I was very happy that my baby was fine. They took care of my baby. They tried whatever possible."* (Mother, age 25).

**3.2 Interactions with other parents.** In addition to turning to their partners, family, and relatives for support at the time of admission and during the stay, fellow mothers in NCU provided emotional and material (food, clothes, and other consumables) support which was a great comfort for many.

*"Other mothers are the ones who helped me feel better. When I looked at their babies and they told me that their babies were sicker when they arrived than [at] that time, I felt better."* (mother, age 27).

*"There was another woman next to me, who had also given birth to twins and whose caregiver was her mother. They used to lend me materials and because she also came from far, when they would have brought them food they cooked and shared with me. I did not have people who could bring me food, so I survived because there were people of goodwill at the hospital."* (mother, age 25).

Other mothers also helped explain the situation of the NCU to others when healthcare providers did not:

*"We had to transmit the information to the mothers who were new and did not know these instructions because healthcare providers were not always in [a] good mood of communicating."* (Mother, age 25).

*"There were mothers without caregivers. They go into the eating hall to see if someone can give them food. We do share food with them because one cannot finish it. I have noticed that people help each other."* (Mother, age 28).

## Theme 4. Financial aspects of the care in NCU and support received

Many mothers and fathers reported that the financial burden due to NCU admission was challenging and was a stress for them. They had difficulty paying for many of the NCU requirements such as specific clothing, basins, flasks as well as the hospital fee. Despite all participants having health insurance, many could not afford the co-payment. Two interviewees reported they had to sell a portion of their land to pay for the NCU costs.

*"I was wondering how I would get money to pay the invoice that was 99 times more than my salary. People told me that I must sell the land to pay medical invoices. I was scared. I was wondering how I will survive when I go home because I had already sold my land in order to buy clothes [for the baby]. I was speechless and there were times when I did not have breast milk because of problems."* (mother, age 44).

*"I was stressed because of the 30,000 RWF medical bill for spending a week and three days in the hospital. I was wondering where I would get the money if I had to spend one more week in the hospital. I was confused but I was also ready to sell my house to be able to pay the medical bill."* (Mother, age 34). She eventually sold her land to pay the bills.

Even sometimes when the NCU was not as expensive as expected, the thought of the potential cost caused a lot of stress.

*"When we went to the hospital, we wondered how we could pay the bill. I was afraid that I would not get enough money to pay the medical bills and would be held in custody at the hospital. But, luckily, I was not charged a lot of money because I had health insurance. I did not have to sell any of my goods."* (mother, age 35).

The cost of hospitalization was a major source of stress, and any forms of intermittent support they had received from others, like porridge and food received from the Catholic sisters or university students supporting the hospital were very much appreciated by the parents.

*"Every morning they [healthcare providers] bring the porridge and bread. I cannot blame them for anything. The healthcare providers bring the hungry people food from the Sisters. They do whatever possible."* (mother, age 36).

*"The most important thing that they helped us with was the free porridge in the morning. It was very impactful because a mother cannot breastfeed the baby without eating the porridge. It is the best help they gave us. We ate the porridge happily with the peace of mind."* (mother, age 25).

## Discussion

Parents shared both positive and negative aspects of their experience inside the NCU in rural Rwanda's district hospital. The parents highlighted common experiences and difficulties adapting to the NCU environments both emotionally and physically, and financial barriers but also unique positive features of the NCU in Rwanda such as high levels of trust of parents for the healthcare providers, and uninterrupted access of mothers to their newborns.

The parents in our study shared similar experiences as reported in other previous studies in general–when their newborns were hospitalized, parents were understandably stressed and worried about their newborn's condition [7]. Consistent to other studies, inclusion in the provision of care contributed to parents' comfort, hope and confidence to keep caring for the newborns even after discharge [7, 26, 32]. According to other studies though, parents not only should be involved in the care, but also in the decision making about the newborn's care [33, 34]. However, in our study, parents did not complain about not being involved in the decision-making for their newborns' care. On the contrary, many placed high trust in their healthcare providers. This could be due to many reasons. Parents may not feel sufficiently informed to make the decision, hence leaving it up to the healthcare providers to decide what is best for the newborn. The parents reported that they trusted the healthcare providers were providing the best possible treatment. There is a need for further research on the parental involvement in decision making in the NCU and other healthcare settings.

Having non-restrictive access to their newborns was one of the positive experiences for mothers. Such access, however, is not always allowed in many NCUs [23] and even other NCUs in Rwanda have restricted access [35]. Limited access to the NCUs for fathers and other family members or caregivers was a source of concern for the parents in our study. Restricted

NCU access is usually justified as necessary for infection control, however, studies present controversial evidence. One study in the United Kingdom found that restricted access was associated with a significant decline in respiratory infections [36], while another study in India showed that allowing parents in the NCU to directly participate in their newborn's care did not increase in hospital acquired infections [37]. NCU access restrictions have particularly been adopted during the COVID-19 pandemic where Neonatal Intensive Care Units that preserved the 24/7 parental presence decreased from 83 to 53% according to a global survey [38]. In all cases, education on hygiene practices are essential for all parents, visitors and providers. In the instances when the parents' full access to the NCU is scientifically justified as harmful to the newborn's health, an open and ongoing communication, emotional support, and the discussion on keeping the newborn connected to parents is needed [39].

Little involvement of fathers in newborn care has particularly been a source of negative experience for mothers and fathers. The lack of father involvement in the newborn care could reduce father's role to merely providing financial support. Studies have shown that little or no involvement of fathers in NCU caused the fathers to be scared of their preterm newborns, affected their early bonding, and eventually father-child relationships [3, 40]. Recognizing the importance of parental access to newborns, several healthcare providers break the rules and allow occasional visitations–leading to reports of "good nurses" and "bad nurses" [41]. Hospitals should reconsider the NCU visitation policy, and aim to deliver family-centered care to promote the best experience for the families and their newborns.

The quality of communication between caregivers and providers greatly impacted the experience of parents in the NCU. When NCU staff provided information about the newborn's condition and showed a caring attitude, parents felt more hopeful, understanding, confident and a will to trust healthcare providers. Similar to the literature, our study highlighted that poor communication and blame by healthcare providers are a source of stress for parents [42].

The parental interactions and peer support were a source of positive experience for parents and should be encouraged. A systematic review has shown that informal or formalized peer support could improve the experiences and well-being of parents [43]. Such findings further reinforced the importance of clear communication from healthcare providers. When healthcare providers gave parents proper and clear orientation and instructions, such messages will be passed along among other parents.

Almost all respondents in our study mentioned the cost was a great source of stress. In Rwanda, 78.5% of the population were enrolled in the Community Based Health Insurance (CBHI) in 2019 [44]. Basic drugs and medical services are covered by CBHI, yet, patients or their families are often required to pay for the more specialized services and diagnostics. Another challenge that often occurs is the medication stock out which requires patients to purchase these at the private pharmacies outside the hospitals hence contributing to higher financial burden. Indirect costs of medical care such as transportation to the health facilities, food, and time away from work are the additional significant financial strains to the families. Similar to the other studies, the cost associated with the NCU care, materials needed for the care, transportation, and food during hospitalization are a serious challenge [45]. There is a need for studies to investigate the entire cost related to the NCU admission, and the related consequences on the families. Such studies would inform policy makers to advocate for expanding insurance coverage and possibly, the deployment of additional support to families with newborns admitted to the NCUs.

## Limitations

This study has some limitations. Most respondents in our study were women, and insights from fathers were relatively limited. The current study was conducted in a district hospital that

receives support from several non-government organizations. The findings may not be transferable to the Rwandan hospitals that do not have such support systems.

## Conclusion

The admission of a newborn into the NCU is a source of stress for parents. Their stress is exacerbated by the environment including barriers to access by fathers and other family members, high cost of neonatal care, and negative interactions with healthcare providers. However, the experience of care is improved by unrestricted access to the newborns by mothers as well as good communication with and trust for healthcare providers. Health facilities should also consider expanding the NCU access, encouraging peer support and actively involving parents in the care of their newborns. Further work is needed to understand the financial barriers for accessing NCU care.

## Supporting information

**S1 File. Semi-structured interview guide (English).**
(DOCX)

**S2 File. Semi-structured interview guide (Kinyarwanda).**
(DOCX)

## Acknowledgments

The authors are grateful to Ruli District hospital administration and Neonatal Care Unit staff, for their support. We thank the parents for taking the time to share their story.

## Author Contributions

**Conceptualization:** Samuel Byiringiro, Catherine M. Kirk.

**Data curation:** Samuel Byiringiro, Catherine M. Kirk.

**Formal analysis:** Samuel Byiringiro, Catherine M. Kirk.

**Funding acquisition:** Samuel Byiringiro, Catherine M. Kirk.

**Investigation:** Catherine M. Kirk.

**Methodology:** Samuel Byiringiro, Deogratias Kaneza, Catherine M. Kirk.

**Project administration:** Samuel Byiringiro.

**Resources:** Deogratias Kaneza.

**Software:** Jenae Logan.

**Supervision:** Rex Wong, Catherine M. Kirk.

**Validation:** Samuel Byiringiro, Catherine M. Kirk.

**Visualization:** Samuel Byiringiro.

**Writing – original draft:** Samuel Byiringiro, Rex Wong, Catherine M. Kirk.

**Writing – review & editing:** Samuel Byiringiro, Rex Wong, Jenae Logan, Deogratias Kaneza, Joseph Gitera, Sharon Umutesi, Catherine M. Kirk.

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
