## [Decision Letter · Decision Letter 0]

1 Feb 2021

PONE-D-20-32408

A Qualitative study to explore the experience of parents of newborns admitted to neonatal care unit in rural Rwanda

PLOS ONE

Dear Dr. Byiringiro,

Thank you for submitting your manuscript to PLOS ONE. After careful consideration, we feel that it has merit but does not fully meet PLOS ONE’s publication criteria as it currently stands. Therefore, we invite you to submit a revised version of the manuscript that addresses the points raised during the review process.

We look forward to receiving your revised manuscript.

Kind regards,

Manuel Fernández-Alcántara, Ph.D.

Academic Editor

PLOS ONE

Additional Editor Comments :

The article present important and interesting data that will be useful for researchers and clinicians. However there are important methodological flaws that author should clarify to consider publication in PlosOne. Please take in consideration the commentaries of reviewer 2 and I also strongly recommend to use the COREQ checklist for reporting of qualitative studies. Please also justify and make a major revision of the results section of the manuscript.

Journal Requirements:

3. In your Methods section, please provide additional information about the participant recruitment method and the demographic details of your participants. Please ensure you have provided sufficient details to replicate the analyses such as:

- the recruitment date range (month and year)

- a description of any inclusion/exclusion criteria that were applied to participant recruitment

- a table of relevant demographic details

- a statement as to whether your sample can be considered representative of a larger population

- a description of how participants were recruited

- descriptions of where participants were recruited and where the research took place.

4. Please provide additional details regarding participant consent. In the ethics statement in the Methods and online submission information, please ensure that you have specified whether consent was informed.

7. Please include your tables as part of your main manuscript and remove the individual files. Please note that supplementary tables should be uploaded as separate "supporting information" files.

Reviewers' comments:

Reviewer's Responses to Questions

**Comments to the Author**

1. Is the manuscript technically sound, and do the data support the conclusions?

Reviewer #1: Yes

Reviewer #2: Partly

2. Has the statistical analysis been performed appropriately and rigorously? 

Reviewer #1: Yes

Reviewer #2: N/A

3. Have the authors made all data underlying the findings in their manuscript fully available?

Reviewer #1: Yes

Reviewer #2: Yes

4. Is the manuscript presented in an intelligible fashion and written in standard English?

Reviewer #1: Yes

Reviewer #2: No

5. Review Comments to the Author

Reviewer #1: This study is very important nowadays. The new philosophy of care involving parents and families is expressed in this manuscript which aim was to determine the family environment, parental stressors in the parents of premature infants in NICU. The study is clear, well organized and results revealed that family environment and stress in parents of premature infants was at the moderate level. Authors focused in the healthcare team that must pay attention to parents preventing the adverse effects of stressors on parents and ultimately on their babies.

Educational and counseling interventions by NICU nurses improves parents' ability to use strategies to manage stress.

The authors should include in the list of abbreviations all abbreviations of the manuscript, namely PTSD

Reviewer #2: Design

This was not a phenomenological design – no theoretical underpinning of the study evidenced. Suggest that this is changed to a qualitative descriptive and contextual study. The design refers to semi-structured interviews, however in the abstract it states: “In-depth interviews

were conducted using a semi-structured guide”.

There is debate about the different types of interviews and how these are categorized. Without having access to the interview guide, it is not clear whether these were semi structured interviews using a pre-determined set of questions or in-depth interviews, with a broad opening question followed by sub questions or prompts. I suggest that this be changed to semi-structured interviews which is appropriate for this type of study.

Participants

Explain the purpose of the stratification of the sample. I assume this was to ensure maximum diversity. What is the rationale for the choice of 3 days in NCU?

Data collection

With more than one interviewer, how was data saturation determined? Was this at data analysis stage? Please clarify.

Data analysis

The authors state that a code book was created based on literature but provide no further substantiation of this process. It would appear that the analysis was deductive, not a phenomenological tradition.

Trustworthiness

Trustworthiness/rigor is not explicitly discussed. Please include this in the revision – e.g., confirmability, audit trail, transferability etc.

The authors should use the COREQ or similar criteria for reporting of qualitative studies.

Ethics

Very limited ethics information provided. Was what done to protect participants’ confidentiality and anonymity? How was risk to the vulnerable participants managed?

Results/findings

The four themes are presented but it is not clear how these emerged – an example of the subthemes and codes need to be provided as evidence of the process. Please explain health insurance in the Rwandan context. The four themes are descriptive, and appropriate to the aim of the study. They are supported by fairly long quotes, but insufficient ‘thick description” is provided. It is preferable to add the participant number as well as description. This section needs rewriting to provide evidence of depth of analysis.

Discussion

Basic discussion covers the most relevant aspect but fails to really engage with literature

Limitations

No limitations included

Conclusion

Recommendations are general and not linked directly to experiences. An example of this is the issue of financial stress and the accompanying recommendations. The is no doubt that the experience of struggling to afford care is a very real one for parents. An explanation of the health insurance structure in Rwanda is needed to contextualize this recommendation

Some language issues which need to be corrected and one or two typographical errors.

6. PLOS authors have the option to publish the peer review history of their article (what does this mean?). If published, this will include your full peer review and any attached files.

Reviewer #1: No

Reviewer #2: No

---

## [Author Response · Author response to Decision Letter 0]

14 Mar 2021

Dear PlosONE Editor,

We appreciated you and the reviewers taking the time to thoroughly review this manuscript and provide us the useful feedback. Please find the point-to-point feedback to the comments made on the manuscript. 

Comments are in black and the feedback in a different color.

The article present important and interesting data that will be useful for researchers and clinicians. However there are important methodological flaws that author should clarify to consider publication in PlosOne. Please take in consideration the commentaries of reviewer 2 and I also strongly recommend to use the COREQ checklist for reporting of qualitative studies. Please also justify and make a major revision of the results section of the manuscript.

Thank you for this feedback. We have included the COREQ checklist (on the last page of this document) with our response to comments. Other feedback to the comments can be found down below.

1.Please ensure that your manuscript meets PLOS ONE's style requirements, including those for file naming. The PLOS ONE style templates can be found at https://journals.plos.org/plosone/s/file?id=wjVg/PLOSOne_formatting_sample_main_body.pdf and https://journals.plos.org/plosone/s/file?id=ba62/PLOSOne_formatting_sample_title_authors_affiliations.pdf

 We have updated the formatting throughout the manuscript to comply with the guidelines of PLOS ONE.

2.Please include additional information regarding the survey or questionnaire used in the study and ensure that you have provided sufficient details that others could replicate the analyses. For instance, if you developed a questionnaire as part of this study and it is not under a copyright more restrictive than CC-BY, please include a copy, in both the original language and English, as Supporting Information.

 A semi-structured in-depth interview guide was developed for this study. We have attached the Kinyarwanda (the local language) and English versions of the semi-structured interview guide as a supplementary material. 

3.In your Methods section, please provide additional information about the participant recruitment method and the demographic details of your participants. Please ensure you have provided sufficient details to replicate the analyses such as:

- the recruitment date range (month and year)

- a description of any inclusion/exclusion criteria that were applied to participant recruitment

- a table of relevant demographic details

- a statement as to whether your sample can be considered representative of a larger population

- a description of how participants were recruited

- descriptions of where participants were recruited and where the research took place.

a.Thank you for the suggestion. We have added the recruitment date and inclusion and exclusion criteria in our methodology section as recommended. Now it read as (p.8): 

Parents or primary caregivers of neonates discharged from the NCU between September 2018 and January 2019 (three months prior to data collection) were included in the study. To ensure that parents and caregivers had been amply exposed to the NCU experience, we only included those whose newborns were admitted to the NCU for at least three days. While some mothers and caregivers were aged below 18 years, we excluded them since their experience tends to be unique because of stigma associated with teenage pregnancy in Rwanda [31]. The acquisition of consent for the participants aged below 18 is additionally different. To ensure diversity and a mix of experiences, we purposively stratified the samples by distance (living within and more than 1-hour walking distance from the facility), by admitting diagnosis (prematurity and other reasons), and outcome (died or alive at discharge). 

b.We have also inserted a paragraph on recruitment procedure to describe how and where participants were recruited (P. 9): 

The recruitment took place between December 2018 and January 2019. After acquiring the approval to access data from Ruli District Hospital leadership, we made the list of eligible potential participants and their geographical locations from the NCU registry. We contacted the nearby Community Health Worker (CHW) to request permission on our behalf and book us an appointment to the potential participants’ homes. On the appointment date, the interviewer was accompanied by the CHW to the potential participant’s home which is where the data collection took place.

c. We have included the table of demographics

d.We have made clear the issue of representativeness among our limitations. (p.25):

This study has some limitations. Most respondents in our study were women, and insights from fathers were relatively limited. The current study was conducted in a district hospital that receives support from a few non-government organizations. Not all Rwandan hospitals receive such support, thus the findings may not be generalizable to other Rwandan health facilities.

4.Please provide additional details regarding participant consent. In the ethics statement in the Methods and online submission information, please ensure that you have specified whether consent was informed.

 We have inserted the consent process in the data collection tool and procedure section, now it reads as (P. 9): 

 The study was approved by the Rwanda National Ethics Committee (No. 107/RNEC/2019) and the Institutional Review Board for the University of Global Health Equity. Ruli 

 District Hospital provided approval for NCU data access and participants signed informed consent prior to the beginning of the interviews. To protect the participants’ 

 confidentiality, we de-identified all quotes included in the results by removing the names, and details that could lead to identify the individual participant or their healthcare 

 provider.

5.We note that you have indicated that data from this study are available upon request. PLOS only allows data to be available upon request if there are legal or ethical restrictions on sharing data publicly. For information on unacceptable data access restrictions, please see http://journals.plos.org/plosone/s/data-availability#loc-unacceptable-data-access-restrictions.

a)If there are ethical or legal restrictions on sharing a de-identified data set, please explain them in detail (e.g., data contain potentially identifying or sensitive patient information) and who has imposed them (e.g., an ethics committee). Please also provide contact information for a data access committee, ethics committee, or other institutional body to which data requests may be sent.

b)If there are no restrictions, please upload the minimal anonymized data set necessary to replicate your study findings as either Supporting Information files or to a stable, public repository and provide us with the relevant URLs, DOIs, or accession numbers. Please see http://www.bmj.com/content/340/bmj.c181.long for guidelines on how to de-identify and prepare clinical data for publication. For a list of acceptable repositories, please see http://journals.plos.org/plosone/s/data-availability#loc-recommended-repositories. We will update your Data Availability statement on your behalf to reflect the information you provide.

 The interview transcripts are all available should they be needed.

6.Your ethics statement should only appear in the Methods section of your manuscript. If your ethics statement is written in any section besides the Methods, please delete it from any other section.

The ethics statement is in the method section (P.9). 

 The study was approved by the Rwanda National Ethics Committee (No. 107/RNEC/2019) and the Institutional Review Board for the University of Global Health Equity. Ruli District Hospital provided approval for NCU data access and participants signed informed consent prior to the beginning of the interviews. To protect the participants’ confidentiality, we utilized participant numbers in the transcripts instead of their names, and we de-identified all quotes included in the results by removing details that could lead to the recognition of the individual participant or their healthcare provider.

7.Please include your tables as part of your main manuscript and remove the individual files. Please note that supplementary tables should be uploaded as separate "supporting information" files.

We have placed all of the tables within the text where they are referenced.

---

## [Decision Letter · Decision Letter 1]

13 Apr 2021

PONE-D-20-32408R1

A qualitative study to explore the experience of parents of newborns admitted to neonatal care unit in rural Rwanda

PLOS ONE

Dear Dr. Byiringiro,

Thank you for submitting your manuscript to PLOS ONE. After careful consideration, we feel that it has merit but does not fully meet PLOS ONE’s publication criteria as it currently stands. Therefore, we invite you to submit a revised version of the manuscript that addresses the points raised during the review process.

Authors have improved the original manuscript. However, further language editing of the manuscript is required. Please include in the data analysis section a discussion about rigor and add the COREQ checklist as a file for reviewers. After these minor revisions the article could be published in PlosOne.

We look forward to receiving your revised manuscript.

Kind regards,

Manuel Fernández-Alcántara, Ph.D.

Academic Editor

PLOS ONE

Journal Requirements:

Reviewers' comments:

Reviewer's Responses to Questions

**Comments to the Author**

1. If the authors have adequately addressed your comments raised in a previous round of review and you feel that this manuscript is now acceptable for publication, you may indicate that here to bypass the “Comments to the Author” section, enter your conflict of interest statement in the “Confidential to Editor” section, and submit your "Accept" recommendation.

Reviewer #1: All comments have been addressed

Reviewer #2: (No Response)

2. Is the manuscript technically sound, and do the data support the conclusions?

Reviewer #1: Yes

Reviewer #2: Partly

3. Has the statistical analysis been performed appropriately and rigorously? 

Reviewer #1: N/A

Reviewer #2: N/A

4. Have the authors made all data underlying the findings in their manuscript fully available?

Reviewer #1: Yes

Reviewer #2: Yes

5. Is the manuscript presented in an intelligible fashion and written in standard English?

Reviewer #1: Yes

Reviewer #2: Yes

6. Review Comments to the Author

Reviewer #1: The manuscript is well written, in a standard English, methods well designed, and the data support the conclusions.

Reviewer #2: Overall the paper has been much improved. It still requires further language editing.Trustworthiness/rigor is still not discussed. I could not find the COREQ list.The authors have still written from a quantitative perspective - e.g. they refer to generalisability which would not apply in a small qualitative study.

7. PLOS authors have the option to publish the peer review history of their article (what does this mean?). If published, this will include your full peer review and any attached files.

Reviewer #1: No

Reviewer #2: No

---

## [Author Response · Author response to Decision Letter 1]

13 May 2021

Dear Editor,

We have addressed the comments from you and the reviewers as follows:

• Reviewer #2: Overall the paper has been much improved. It still requires further language editing. Trustworthiness/rigor is still not discussed. I could not find the COREQ list.The authors have still written from a quantitative perspective - e.g. they refer to generalisability which would not apply in a small qualitative study.

o We added the rigor/trustworthiness statements in both the analysis and data collection sections. 

o We added the COREQ checklist in the supplementary files

o We changed the generalizability to transferability of the findings

o We did the language editing according to the recommendations given by the editor

• Additional changes

o We removed the ethics statement from the declaration section.

o To answer all relevant questions asked by the COREQ, we added the average duration of interviews under the results section,

o We included a statement which explains why mothers reported tiredness looking after their babies day and night – according to policy at the study site on page 16

o We did language editing on page 25 to clarify why patients still suffer the out of pocket burden while covered by the Community Based Health Insurance.

o Reviewed the references to ensure they are all accessible

---

## [Editor Report · Decision Letter 2]

24 May 2021

A qualitative study to explore the experience of parents of newborns admitted to neonatal care unit in rural Rwanda

PONE-D-20-32408R2

Dear Dr. Byiringiro,

We’re pleased to inform you that your manuscript has been judged scientifically suitable for publication and will be formally accepted for publication once it meets all outstanding technical requirements.

Kind regards,

Manuel Fernández-Alcántara, Ph.D.

Academic Editor

PLOS ONE
---

## [Editor Report · Acceptance letter]

5 Aug 2021

PONE-D-20-32408R2 

A qualitative study to explore the experience of parents of newborns admitted to neonatal care unit in rural Rwanda 

Dear Dr. Byiringiro:

I'm pleased to inform you that your manuscript has been deemed suitable for publication in PLOS ONE. Congratulations! Your manuscript is now with our production department. 

Kind regards, 

on behalf of

Dr. Manuel Fernández-Alcántara 

Academic Editor

PLOS ONE